# Anti-Cancer Activity and Phenolic Content of Extracts Derived from Cypriot Carob (*Ceratonia siliqua* L.) Pods Using Different Solvents

**DOI:** 10.3390/molecules26165017

**Published:** 2021-08-19

**Authors:** Gregoria Gregoriou, Christiana M. Neophytou, Alexandru Vasincu, Yiota Gregoriou, Haria Hadjipakkou, Eftychia Pinakoulaki, Marios C. Christodoulou, Georgia D. Ioannou, Ioannis J. Stavrou, Atalanti Christou, Constantina P. Kapnissi-Christodoulou, Siegfried Aigner, Hermann Stuppner, Antonis Kakas, Andreas I. Constantinou

**Affiliations:** 1Department of Biological Sciences, University of Cyprus, Nicosia 1678, Cyprus; gregoriou.gregoria@ucy.ac.cy (G.G.); gregoriou.panayiota@ucy.ac.cy (Y.G.); andreasc@ucy.ac.cy (A.I.C.); 2European University Research Center, Nicosia 2404, Cyprus; 3Department of Pharmacodynamics and Clinical Pharmacy, Faculty of Pharmacy, “Grigore T. Popa” University of Medicine and Pharmacy, 700115 Iaşi, Romania; alexandru.vasincu@umfiasi.ro; 4Department of Chemistry, University of Cyprus, Nicosia 1678, Cyprus; hadjipakkou.charia@ucy.ac.cy (H.H.); effiep@ucy.ac.cy (E.P.); mchris39@ucy.ac.cy (M.C.C.); gioann02@ucy.ac.cy (G.D.I.); stavrou.ioannis@ucy.ac.cy (I.J.S.); christou.atalanti@ucy.ac.cy (A.C.); ckapni1@ucy.ac.cy (C.P.K.-C.); 5Department of Life Sciences, European University Cyprus, Nicosia 2404, Cyprus; 6Institute of Pharmacy/Pharmacognosy, Center for Molecular Biosciences Innsbruck (CMBI), Center for Chemistry and Biomedicine, University of Innsbruck, 6020 Innsbruck, Austria; si.aigner@tsn.at (S.A.); hermann.stuppner@uibk.ac.at (H.S.); 7Department of Computer Science, University of Cyprus, Nicosia 1678, Cyprus; antonis@cs.ucy.ac.cy

**Keywords:** carobs, chemoprevention, polyphenols, myricetin, naringenin, kaempferol, anti-oxidant activity, apoptosis, LC-MS, antiproliferative capacity

## Abstract

Extracts derived from the *Ceratonia siliqua* L. (carob) tree have been widely studied for their ability to prevent many diseases mainly due to the presence of polyphenolic compounds. In this study, we explored, for the first time, the anti-cancer properties of Cypriot carobs. We produced extracts from ripe and unripe whole carobs, pulp and seeds using solvents with different polarities. We measured the ability of the extracts to inhibit proliferation and induce apoptosis in cancer and normal immortalized breast cells, using the MTT assay, cell cycle analysis and Western Blotting. The extracts’ total polyphenol content and anti-oxidant action was evaluated using the Folin–Ciocalteu method and the DPPH assay. Finally, we used LC-MS analysis to identify and quantify polyphenols in the most effective extracts. Our results demonstrate that the anti-proliferative capacity of carob extracts varied with the stage of carob maturity and the extraction solvent. The Diethyl-ether and Ethyl acetate extracts derived from the ripe whole fruit had high Myricetin content and also displayed specific activity against cancer cells. Their mechanism of action involved caspase-dependent and independent apoptosis. Our results indicate that extracts from Cypriot carobs may have potential uses in the development of nutritional supplements and pharmaceuticals.

## 1. Introduction

*Ceratonia siliqua* L., also known as the Mediterranean carob tree, has been widely studied for the functional chemicals contained in its fruit. The carob fruit contains two major parts (by weight): the pulp (90%) and the seeds (10%). The chemical composition of the carob pulp differs widely depending on cultivar or species, origin, climate, as well as stage of maturity during harvesting [1,2,3]. Recently, research has focused on the many beneficial effects of carobs for human health [4,5]. In addition to carobs, recent evidence suggests that extracts derived from peels [6], leaves [7] and seeds [8] of different plant species, edible or not, may induce diverse health-promoting effects, often different from that expected, due to their rich content of bioactive compounds. The carob fruit is a complex mixture of mainly sugars and fibers, followed by a great diversity of polyphenols. Carob pulp constituents, including fibers, cyclitols, polyphenols and tannins have been associated with the health-promoting effects of carobs against diabetes, diarrhea and cancer [9,10,11,12,13]. Additionally, the anti-cancer properties of carob pulp have been attributed mainly to the presence of polyphenolic compounds, which constitute up to 20% of the carob pulp [14,15].

The ripening of the carob fruit significantly affects its composition in phenolic content and, therefore, its biological properties. Unripe carobs have higher amounts of phenolic acids, polyphenols, flavonoids and tannins and display higher in vitro anti-oxidant capacity whereas the ripe fruit contains high levels of Gallic acid (GA), due to the enzymatic hydrolysis of gallotannins in immature fruit [3]. GA, a phenolic acid, as well as Myricetin, Naringenin and Kaempferol, flavonoids commonly found in carob fruit, have been shown to exert anti-cancer properties in vitro by activating caspase-dependent programmed cell death (CD-PCD) and cell cycle arrest [16,17,18,19,20,21]. Mitochondrial or intrinsic apoptosis is controlled by the members of the Bcl-2 superfamily. Anti-apoptotic Bcl-2 is often overexpressed in cancer cells and blocks mitochondrial membrane pore permeabilization (MOMP). Following apoptosis induction, Bcl-2 is reduced, pro-apoptotic factors exit the mitochondria and activate caspase-9 which cleaves downstream caspases-3, -6 and -7 [22]. In addition to CD-PCD, many natural agents can also cause cell death by caspase-independent apoptosis (CI-PCD), via cleavage of the apoptosis inducing factor (AIF). Even though different proteins are implicated in these two pathways, the crosstalk between them has been reported in the literature [23].

Cyprus is among the top producers of carobs worldwide. The chemical and biological properties of Cypriot carobs and derived products have recently come to focus. Kyriacou et al. revealed that major metabolic events occur during carob ripening that significantly affect the phenolic content and anti-oxidant capacity of carob-pod derived extracts [3]. Kibble size was found to be an important parameter in determining the chemical characteristics of the resulting carob juice produced by different extraction methods [24]. The geographical origin, type and nutritional composition of carobs from Cyprus and other countries, were differentiated based on Fourier transform infrared (FTIR) spectroscopy and chemometrics [25,26]. In addition, the profile of biogenic volatile organic compounds (VOCs) emitted from carob fruit that contribute to its unique aroma, has been recently described [27]. However, the potential anticancer effects of carob fruit extracts have not yet been explored. Our main objectives were to evaluate and compare for the first time, the anti-cancer potential of carob extracts derived from Cypriot carobs at different maturity stages and from different parts of the carob pods. In addition, we assessed the properties of extracts derived in solvents of different polarity, that is known to affect polyphenol content [28]. We investigated the anti-proliferative and pro-apoptotic effect of the carob extracts using established cancer and normal breast cell lines. Furthermore, we determined the total polyphenolic content (TPC) of these extracts and their anti-oxidant ability. Finally, we performed LC-MS and identified the dominant polyphenols in the most potent anti-cancer extracts. Our results indicate that depending on the extraction solvent used and stage of maturity, carob fruit extracts from trees grown in Cyprus have anti-cancer properties that may be attributed to the presence of polyphenols that can induce caspase-dependent and independent apoptosis in breast cancer cells.

## 2. Results

### 2.1. Carob Extracts Significantly Reduce the Viability of MCF-7 Cells

Initially, we obtained extracts of the whole ripe carobs using Diethyl ether (DE), Ethyl acetate (EA), Ethanol (EtOH) and water as solvents. MCF-7 breast cancer and MCF-10A “immortalized” breast cell lines were treated with increasing concentrations of ripe carob extracts for 48 h. The DE and EA extracts were more effective than the EtOH and Aqueous extracts, based on their IC_50_ (Table 1, Figure 1 and Appendix A). Consequently, the DE and EA extracts produced from the whole fruit (pulp + seeds) induced a marked reduction in MCF-7 viability of 50% at a concentration of 0.26 mg/mL and 0.44 mg/mL, respectively. Treatment with the same carob extracts did not show a significant effect in reducing the viability of the normal breast cell line, MCF-10A (Figure 1A,B). The EtOH and Aqueous extracts were also specific in reducing the viability of the MCF-7 cell line at concentrations ranging from 0–2.5 mg/mL (Appendix A).

Among the four solvents used in the extraction process, the DE and EA extracts were found to possess the strongest anti-proliferative effects in established cell lines (Table 1). For this reason, we used the DE and EA to produce separate extracts from the pulp and the seeds of the ripe carobs. The EA extract produced from the pulp of ripe carobs significantly reduced the viability of MCF-7 cells at concentrations starting at 0.5 mg/mL, while it had no significant effect on the normal MCF-10A cell line (Figure 1C). The DE extract from the ripe carob pulp also showed specificity for the cancer cell line. (Figure 1D); the DE ripe seed extract had no significant effect on both cell lines (Figure 1F) while the EA ripe seed extract reduced the viability of both cell lines at selected concentrations (Figure 1E). The IC_50_ values for the “seeds” and “pulp” extracts are shown in Table 2.

### 2.2. Anti-Proliferative Effects of Unripe Carob Extracts

For subsequent experiments we focused on DE and EA, the solvents that produced the most potent and selective extracts in ripe carobs. Thus, extracts were prepared using separately the pulp, seeds and the whole unripe fruit and their effects on cell proliferation were examined. As shown in Table 3 and Appendix A, EA extracts made from the whole unripe carobs had the most potent anti-proliferative effect on MCF-7 cells with an IC_50_ of 0.82 ± 0.08 mg/mL. Previous studies have reported that the pulp contains higher amounts of polyphenols compared to seeds [4]. In fact, treatment of the cell lines with the EA and DE extracts made from unripe seeds did not demonstrate any significant effects in terms of cell viability (Table 3 and Appendix A). In contrast, the viability of both MCF-7 and MCF-10A cells decreased with increasing concentrations of the unripe pulp EA and DE extracts at concentrations ranging from 0.1 to 0.25 mg/mL (Table 3 and Appendix A), while the EA unripe pulp extract was effective in reducing the proliferation of both cell lines at lower concentrations (Appendix A). The EA extract produced using both pulp and seeds of unripe carobs was specific in reducing the viability of MCF-7 cells only (Table 3 and Appendix A). In contrast, the DE extract produced from the whole unripe carobs reduced the viability of both MCF-7 and MCF-10A cell lines with an IC_50_ of 0.42 ± 0.20 mg/mL and 1.75 ± 0.19 mg/mL, respectively (Table 3 and Appendix A).

### 2.3. Cell Cycle Analysis of the Ripe and Unripe Carob Extracts

The effects of the extracts from ripe carobs on cell cycle distribution of MCF-7 and MCF-10A cells were evaluated using PI staining as described previously [29]. Briefly, cells were incubated with selected concentrations of DE and EA ripe carob extracts as indicated (Figure 2). The DE and EA ripe pulp + seeds or ripe pulp extracts caused an increase in the subG1 phase which is indicative of apoptosis. Importantly, these treatments did not have a significant effect on the normal breast cells (Figure 2A–D), with the exception of the EA ripe pulp extract at 0.25 mg/mL (Figure 2C). Furthermore, treatment of the cells with the EA ripe seeds extract caused an increase of the subG1 phase specifically in the MCF-7 cell line (Figure 2E). On the other hand, no significant effects were observed using the DE ripe seeds on both cell lines (Figure 2F).

To determine whether the growth inhibitory effects observed using the unripe carob extracts were due to cell cycle arrest, we analyzed the effects of the unripe carob extracts on cell cycle distribution. Cells were incubated with different concentrations of unripe carob extracts, and the results are illustrated in Figure 3. Treatment of cells with the EA unripe pulp + seeds extract at the concentration of 0.75 mg/mL increased the percentage of MCF-7 cells in the SubG1 phase to 23% compared to 2.7% seen in the control. Importantly, it did not exert any significant changes in the cell cycle distribution of the MCF-10A cell line compared to the control (Figure 3A). This supports the results observed using the MTT assay, in which we found that the EA extracts made from the whole unripe carobs had a potent anti-proliferative effect on MCF-7 cells with no significant effects on the MCF-10A cell line (Appendix A).

Treatment with the EA unripe pulp extract at a concentration of 0.25 and 0.50 mg/mL increased the percentage of MCF-10A cells in the G1 phase from 42.8% (control) to 65.4% and 63.1%, respectively. At 0.50 mg/mL, the EA unripe pulp extract increased the fraction of MCF-7 cells to 20% in the S phase compared to the untreated cells where the percentage of cells was 11.5% (Figure 3C). In both cell lines, the EA unripe pulp induced a moderate increase in the SubG1 population (around 5%). This is in agreement with the MTT assay where treatment of the cell lines with the EA unripe pulp extracts at concentrations ranging from 0.1 to 0.50 mg/mL decreased the viable cell number of both MCF-7 and MCF-10A cell lines (Appendix A). Treatment of the cells with the EA unripe seeds did not show any significant effects on either cell line (Figure 3E and Appendix A).

Incubation with the DE unripe pulp + seeds at 0.25 and 0.50 mg/mL induced G1 arrest in MCF-10A cells by increasing the G0/G1 fraction by 18.5% and 23.8%, respectively, while it also caused a subG1 increase in MCF-7 cells by 10% (Figure 3B). The DE unripe pulp extract at 0.25 and 0.50 mg/mL caused moderate increases in the subG1 and G1 fractions of MCF-10A cells and a 10% increase in the subG1 phase of MCF-7 cells (Figure 3D). Incubation with the DE unripe seeds increased the percentage of MCF-7 cells in the SubG1 phase from 3.1%, as seen in the control, to 24.61% at the concentration of 0.50 mg/mL (Figure 3F).

### 2.4. Carob Extracts Promote Apoptosis through the Caspase-Dependent and Independent Pathways

To further investigate the underlying anti-cancer mechanism of carob extracts, we chose the extracts that displayed the highest cytotoxicity selectively against the MCF-7 cell line, in both the MTT assay and in cell cycle distribution. Thus, we used the “ripe pulp + seeds” and “ripe pulp” extracted with DE or EA and the EA unripe pulp + seeds extract and performed Western blot analysis.

We evaluated the apoptotic signaling pathway induced by the Carob extracts in MCF-7 cells by monitoring the expression status of proteins involved in the caspase-dependent or independent pathways. Since MCF-7 cells are caspase-3 deficient [30] we measured the levels of effector caspase-7. Both the DE “ripe pulp + seeds”, and DE “ripe pulp” extracts induced cleavage of initiator caspase-9 and induced proteolysis of its downstream targets, caspase-7 and caspase-6. Both extracts triggered the cleavage of α-Fodrin and PARP which are known substrates of caspase-3 and caspase-7 during the early stages of apoptosis [31]. The levels of Bcl-2, a protein controlling mitochondrial apoptosis as well as AIF, a caspase-independent pro-apoptotic factor released from the mitochondria, were reduced in the presence of DE Ripe pulp (Figure 4B). This indicated the activation of both CD-PCD and CI-PCD by the DE extracts. In addition, both extracts decreased the phosphorylation of AKT, blocking survival signals, and increased the levels of cell cycle inhibitors p21 and p27.

α-Fodrin proteolysis to its cleaved form (150 kDa) was evident following treatment with the EA ripe pulp + seeds extracts (Figure 5A). Incubation with the EA “ripe pulp + seeds” and “ripe pulp” extracts resulted in a significant decrease in the protein levels of the anti-apoptotic protein Bcl-2 and in increased levels of the CDK inhibitors p21 and p27 (Figure 5A,B). AIF was cleaved following treatment with the EA “ripe pulp + seeds” but its levels increased the EA “ripe pulp”. This could be explained due to the accumulation of nuclear AIF prior to its cleavage and, therefore, apoptosis induction. The phosphorylation of AKT was not significantly affected by the EA extracts (data not shown).

The extract produced by the unripe fruit using EA provided similar results as the ripe extracts. The EA unripe pulp + seeds extract induced cleavage of caspases-7 and -9 as well as PARP proteolysis (Figure 6A,B). Caspase-9 cleavage was observed by the increase of the p17 subunit in all cases, which represents the active fragment without the CARD pro-domain [32,33]. In addition, the EA unripe pulp + seeds extract reduced AIF levels and increased p21 and p27 protein expression (Figure 6A,C).

### 2.5. Total Phenolic Composition and Anti-Oxidant Activity of Carob Extracts

The extracts made from the ripe and unripe carobs extracted with DE and EA were evaluated for their total phenolic composition (TPC) as well as their anti-oxidant activity as described in the methods section. The TPC of these extracts ranged from 6.20 to 140.41 mg/g of extract (Table 4), with the exception of the extracts made from the DE ripe and unripe seeds where no phenolic content was detected. As shown in Table 4, the EA unripe pulp + seeds displayed the highest total phenolic content of 140.41 mg/g with an EC_50_ of 11.51 μg/mL. The DE ripe pulp and EA ripe pulp extracts showed a lower TPC of 6.20 and 11.57 mg/g with an EC_50_ of 1618.7 and 165.08 μg/mL, respectively. Overall, the extracts that exhibited the lowest EC_50_ (and, therefore, strongest anti-oxidant capacity) had the highest TPC.

### 2.6. Composition of the Extracts with Potent Anti-Cancer Effects Based on LC-MS Analysis

LC-MS analysis of DE and EA ripe pulp + seeds carob extract was performed in order to detect and quantify phenolic compounds (Table 5). Twelve major polyphenols were examined, and only eight of them were detected in the examined extracts. Quercitrin, Caffeic acid, Sinapic acid and Catechin were not detected compared to Apigenin, Myricetin, Rutin, Naringenin, Ferulic acid, Kaempferol, Gallic acid and Quercetin. In the case of Quercetin, quantification was not possible due to coelution with other substances with the same *m/z* ratio. The analyte with the highest concentration in both samples was Myricetin with 40.67 mg/mL in EA extract and 53.76 mg/mL in DE extract. It is worth to mention here that in the DE extract, Kaempferol was approximately 5 times the concentration of that in EA. Furthermore, Ferulic acid in DE was detected but not quantified as the concentration was found lower than LOQ. All the concentrations of quantified substances are shown in Table 5. It was also observed that the use of DE extraction procedure resulted in higher concentrations of phenolic compounds than in the case of EA. The results of chromatograms of EA and DE are represented in Appendix A, respectively. The retention times are shown in Appendix A.

## 3. Discussion

As part of this study, we obtained extracts from the whole carob fruit, as well as the seeds and pulp separately. We investigated the anti-proliferative effects of Cypriot carob extracts in cancer cells and normal breast cell lines, their ability to induce apoptosis as well as the mechanism by which they exert their action. In addition, we determined the TPC of each extract as well as their anti-oxidant activity. Finally, we performed LC-MS analysis in the most promising anti-cancer extracts. Although several studies have been performed for the identification and quantification of polyphenols in carob fruit, research has highlighted that different extraction methods produce diverse patterns and compositions, changing the phenolic profile, and, therefore, the function of the derived extract [24,34].

The DE extract from the “ripe pulp + seeds” had the lowest IC_50_ and was selective for inhibiting the proliferation of cancer cells (Table 1 and Figure 1B). Even though the DE extract from the unripe “pulp + seeds” exerted high cell growth inhibition on the MCF-7 cell line, it also inhibited the growth of the normal MCF-10A cells (Table 3 and Appendix A). The reason behind this may be due to the variations in phenolic content based on the carob’s ripening stage [35]. Research studies have reported that during maturity, the total phenolic content varies. The unripe carobs contain higher total amount of polyphenols, but the ripe carobs contain the highest amounts of GA, possibly due to the degradation of gallotannins [3,35]. In fact, in this study, it was demonstrated that the highest TPC was found in extracts made using the unripe carobs (Table 4). The DE ripe pulp had low TPC but was relatively selective for cancer cells (Table 4 and Figure 1D). The extracts made from the DE ripe seeds did not exert any anti-proliferative effects on either cell line. This is in agreement with our results from the TPC analysis (Table 4) and other research studies, which reported that the carob seeds have lower amounts of polyphenols compared to the pulp [36].

On the other hand, the EA extracts obtained from the “ripe” and “unripe pulp + seeds” both selectively reduced the viability of the MCF-7 cell line at the concentrations tested and also displayed relatively high TPC (Figure 1A, Appendix A and Table 4). The EA ripe seeds extract displayed a TPC of 14.74 mg/g but was not selective for either cell line while the EA ripe pulp had similar TPC content (11.57 mg/g) but was efficient only against the cancer cells. Finally, the EA ripe seeds had no anti-proliferative effect and insignificant TPC (Appendix A and Table 4). Taken together, the current findings demonstrate the dose-dependent anti-proliferative impacts of the extracts examined, with the most potent being the “Ripe Pulp + Seeds” extracted with DE, followed by “Ripe Pulp” extracted with EA and the “Unripe Pulp + Seeds” extracted with EA. Based on the US National Cancer Institute (NCI) guidelines, a crude extract is generally considered to have high in vitro cytotoxic activity if the IC_50_ value is ≤ 30 μg/mL [37,38]. Pezzuto et al. consider extracts with higher IC_50_ values (100 μg/mL) to be cytotoxic and eligible candidates for further studies [39], while several recently published papers [40,41,42,43,44,45] explore the biological activity of plant extracts with IC_50_ values higher than 200 μg/mL. Importantly, all the extracts made from the whole fruit using EA and DE induced apoptosis in MCF-7 cells with no significant effects on the normal breast cell line MCF-10A. This was also evident by the ability of these extracts to increase the subG1 fraction selectively in MCF-7 cells as evaluated with flow cytometry (Figure 2A,B). The presence of the subG1 fraction is indicative of apoptosis [46,47]. Interestingly, in some cases, apoptosis induction was measured around 20% (Figure 2), while cell viability as measured by the MTT assay, decreased by almost 80% (Figure 1). This indicates that other forms of cell death, such as necrosis, may contribute to the observed anti-cancer effects of the carob extracts; Annexin V/Propidium iodide staining followed by flow cytometry as well as cell morphology as observed by microscopy is necessary to distinguish between apoptotic and necrotic death potentially induced by effective extracts. In addition, even though the EA extract from ripe pulp displayed potent anti-proliferative capacity (Figure 1C, Table 2), it also increased the subG1 fraction of normal cells (Figure 2C). Future studies should include the sub-fractionation of crude carob extracts in an effort to improve their IC_50_ values against cancer cells and screening of the crude extracts and their sub-fractions against a panel of cancer and normal cell lines to evaluate their effectiveness in a range of different cancer types.

Interestingly, the EA ripe seeds extract caused a significant increase of the subG1 phase selectively in cancer cells, revealing that it may have anti-cancer properties (Figure 2E). Recent reports suggest that carob seed extracts could potentially be a source of bioactive anti-oxidant compounds [48]. Even though the seeds, leaves and roots of different plant species have been mainly considered as food waste for years, they are now receiving attention for their bioactive constituents and their potential uses in the production of dietary supplements [7,49] or as sources of plant biostimulants [50,51]. Further characterization of the chemical composition of extracts derived from waste plant matrices is needed to reveal their full potential as possible sources of functional compounds [52].

To evaluate the anti-oxidant activity of carob extracts, the DE and EA extracts were further analyzed using the DPPH assay. The most effective anti-oxidant extract was found to be the EA unripe pulp + seeds, with an EC_50_ of 11.51 μg/mL and a TPC of 140.41 mg/g as determined using the Folin–Ciocalteu method. This is in agreement with a recent study by Kyriacou et. al, that highlighted the anti-oxidant potential of immature carob fruits [3]. The study reported that during maturity, there was a swift decline in the fruit total phenolic content, catechins, tannins and flavonol glycosides, which resulted in the loss anti-oxidant capacity from the ripe fruit pulp extracts.

To investigate the mechanism of cell death induced by the carob extracts, we monitored the potential activation of key proteins involved in major apoptotic pathways. Caspase-9 functions in the intrinsic pathway of apoptosis and it is activated following cytochrome c release from the mitochondria. Once Caspase-9 is activated it can trigger the activation of the effector caspases -3, -6 and -7. Subsequently, effector caspases can cleave various substrates such as PARP and α-Fodrin resulting in cell disruption and eventually cell death [22]. Caspase-7 takes on the role of the main effector caspase, when caspase-3 is missing [53]. In caspase-3 deficient MCF-7 cells, we show that, DE “ripe pulp + seeds” and “ripe pulp” treatment induces the proteolysis of α-Fodrin accompanied by a decrease of the full-length protein expression of Caspase-7 (Figure 4 and Figure 5). PARP expression was decreased following treatment with all of the carob extracts at selected concentrations. Additionally, the EA unripe pulp + seeds, DE “ripe pulp + seeds” and “ripe pulp” extracts induced the cleavage of caspase-9. Importantly, Bcl-2, an anti-apoptotic protein, was decreased following treatment with the DE ripe pulp, EA “ripe pulp + seeds” and “ripe pulp” extracts. However, Bcl-2 expression did not decrease following treatment with the EA unripe pulp + seeds (data not shown) and DE ripe pulp + seeds extracts, suggesting that other protein members of the Bcl-2 family may be involved in inducing MOMP.

Following treatment with the carob extracts, AIF was decreased from its 67 kDa form which indicates its proteolysis to its soluble form (57 kDa), with the exception of EA ripe pulp treatment which resulted in increased protein levels of AIF. This indicates the accumulation and subsequent proteolysis of AIF to its soluble form and the involvement of the caspase-independent mechanism of apoptosis [54]. In addition, we showed that the DE carob extracts inhibited the activation of the AKT pathway. Enhanced activation of AKT which occurs mainly via the phosphorylation at Ser473, leads to suppression of apoptosis [55,56,57]. The decrease in AKT phosphorylation may explain the observed apoptosis induced by the carob extracts. The cell cycle inhibitors, p21 and p27, that are targets of the AKT pathway, were found to be upregulated following treatment with all of the carob extracts, indicative of inhibition of proliferation [58].

Carob polyphenols have attracted scientific interest due to their health promoting effects. GA, the most abundant phenolic acid found in carobs, displays anti-cancer effects by inducing cell cycle arrest, and apoptosis via activating the caspase-dependent pathway and ROS generation [59]. Previous studies have reported that GA was found at the highest concentration in the pulp compared to seeds. Indeed, the total phenolic content of GA in the pulp has been estimated to be between 0.45 and 53.7 mg equivalents/g of dry extract [4] while in the seeds it has been estimated to be 0.19 mg equivalents/g of dry extract [36,60]. Based on the LC-MS analysis (Table 5), Myricetin had the highest concentration in both tested samples from the ripe whole fruit; the DE extract had the highest levels of Myricetin and also displayed better IC50 values than the EA extract (Table 1). Myricetin has been previously found to induce apoptosis in MCF-7 and triple negative MDA-MB-231 breast cancer cells [61,62]. Naringenin that was also detected in high levels in both extracts, has been found to induce tumor cell death and inhibit angiogenesis in malignant melanoma while it affected inflammatory and apoptosis pathways to inhibit migration of breast cancer cells [63,64]. Interestingly, in the DE extract, Kaempferol was approximately 5 times the concentration of that in EA. Kaempferol has been reported to induce apoptosis in breast cancer cell lines in vitro, through downregulation of Bcl-2 and cleavage of PARP [65,66], similarly to the carob extracts (Figure 4A and Figure 5A). Even though Myricetin, Naringenin and Kaempferol were detected in high levels in the examined extracts and these compounds have been found to induce apoptosis in cancer cells by previous studies, it is not clear to what extend they contribute to the anti-cancer effects observed here. Sub-fractionation of the most potent extracts and thorough investigation of their mechanism of action will reveal whether these effects may be attributed to polyphenols or other constituents.

Overall, our results showed that the anti-proliferative capacity of carob extracts varied with the stage of maturity of carobs and the solvent used for the extraction process. Regarding the ripe carobs, our results demonstrated that the DE and EA whole ripe extracts were highly specific in reducing the viability of MCF-7 cells compared to seeds, pulp, EtOH and Aqueous extracts. Previous studies showed that polyphenols are more soluble in solvents less polar than water [28], suggesting that the DE and EA solvent extraction was able to dissolve higher amounts of polyphenols compared to EtOH and Aqueous extraction. These findings could explain the anti-proliferative effects observed using the DE and EA whole ripe carob extracts compared to the other solvents used.

## 4. Materials and Methods

### 4.1. Plant Material

Carobs were collected from a single tree growing near Delikipos village bearing fruits typical of the predominate local landrace “Tillyria”. Delikipos is a semi-mountainous area (300 m altitude) located in the central part of the island within one of the traditional centers of carob cultivation. Agro-environmental conditions at the Delikipos area favors the production of carobs with higher concentrations of condensed tannins, total phenolics, organic acids and other secondary metabolites compared to carobs produced to the coastal zone of the island [67].

### 4.2. Extraction Methods

The carobs and by-products (seeds and pulp) were cut in small pieces and grounded using a food processor (Thermomix). Next, the samples were immersed in liquid nitrogen and freeze-dried under pressure for 24 h. The lyophilized samples (seeds, pulp and whole carob) were grounded in a food processor to produce a fine powder and sieved through a stainless-steel mesh. Subsequently, 250 g of carob powder obtained from the whole ripe carobs was successively extracted using a series of solvents of increasing polarity (3 × 800 mL, for each solvent for 6 h) under magnetic stirring at room temperature. The order of polarity is as follows: Diethyl ether (DE) < Ethyl acetate (EA) < Ethanol (EtOH) < Water.

For the aqueous extract the procedure was performed once. The supernatants of each solvent were combined and filtered using a Whatman No. 1 filter paper. The Aqueous extract was lyophilized for 24 h while the DE, EA and EtOH extracts were placed into a round bottom flask and evaporated under pressure at 40 °C using the rotary evaporator, lyophilized for 24 h and stored at −20 °C for future use. For the extraction of the pulp, seeds, and pulp + seeds, from the ripe and unripe carobs we used the DE and EA solvents only and as described above (Figure 7).

The extract yields were calculated as follows:Yield (%) = A/B ×100(1)
where A is the weight of the extract and B is the weight of the initial carobs.

The yield of extraction is shown in Table 6.

Among the four solvents used in the extraction process, the DE and EA extracts were found to possess the strongest anti-proliferative effects in established cell lines (Table 1). Based on these results, we used only these two solvents for the extraction of unripe carobs (pulp + seeds, pulp or seeds). The yield of extraction of unripe carobs is shown in Table 7.

### 4.3. Cell Culture and Reagents

MCF-7 and MCF-10A cell lines were obtained from the American Type Culture Collection (ATCC) (Manassas, VA, USA). MCF-7 breast cancer cell line was cultured in DMEM supplemented with 10% Fetal Bovine Serum (FBS) and 1% antibiotic/antimycotic. MCF-10A immortalized breast cell line was cultured in DMEM F12 supplemented with 20 ng/mL EGF, 100 ng/mL Cholera Toxin, 500 ng/mL Hydrocortizone, 10 μg/mL Insulin, 5% Horse Serum (HS) and 1% antibiotic/antimycotic. DMEM, FBS, HS, antibiotic/antimycotic and trypsin were purchased from Gibco, Invitrogen. EA, DE and EtOH used for the preparation of carob extracts were purchased from Sigma-Aldrich (St. Louis, MO, USA). PARP, AKT, p-AKT, Bcl-2, AIF and Caspases-6, -7, -8, -9 antibodies were purchased from Cell Signaling Technology (Danvers, MA, USA). GAPDH, p21, p27 and α-Fodrin antibodies were purchased from Santa Cruz Biotechnology Inc. All polyphenolic standards are analytical grade. Apigenin, Quercetin, Sinapic acid, Caffeic acid, Naringenin and Gallic acid were purchased from Sigma-Aldrich (St. Louis, MO, USA). Catechin, Myricetin and Quercitrin were supplied from HWI ANALYTIK GMBH (Rulzheim, Germany). Kaempferol and Rutin were purchased from LGC Standards GmbH (Luckenwalde, Germany) and PhytoLab GmbH and Co (Vestenbergsgreuth, Germany), respectively. All other reagents were purchased from Sigma Aldrich.

### 4.4. MTT Assay

The MTT assay was performed as described previously [68] with a few modifications. A total of 5 × 10^4^ cells were seeded per well of a 96-well plate. The breast cell lines MCF-7 and MCF-10A were incubated overnight to allow for cell attachment and recovery. Cells were treated with increasing doses of Carob extracts (0.1–2.5 mg/mL) and incubated for 48 h at 37 °C. Cell viability was measured using the MTT 3-(4,5-dimethylthiazol-2-yl)-2,5-mono tetrazolium bromide assay. At the end of each incubation period, 20 μL of MTT dye (1 mg/mL; Sigma St. Louis, MO, USA) was added in each well and the plate was incubated at 37 °C for 4 h. Subsequently, the plates were immediately read on a microplate reader (Wallac, PerkinElmer, MA, USA) at 570 nm. Absorbance was proportional to the number of viable cells per well. Percentage of cell viability in each group was calculated after normalization to its own control.

### 4.5. Cell Cycle Analysis

Following incubation, cells were treated with different concentrations (as described in the figure legends) of carob extracts for 48 h. They were harvested, fixed in 70% ethanol, incubated with the Propidium Iodide (PI) staining solution (containing 1 mg/mL PI and 100 μg/mL Rnase) for 30 min at 37 °C and analyzed for DNA content using the Guava EasyCyte flow cytometer and the GuavaSoft analysis software (Millipore, Watford, UK).

### 4.6. Western Blotting

Following incubation with selected carob extracts at different concentrations (as indicated in the figures), cells were washed with ice-cold PBS and lysed in RIPA buffer (150 mM NaCl, 50 mM Tris, 5 mM EDTA [Na2], 1% (*v*/*v*) Triton X-100, 1% (*w*/*v*) deoxycholate (24 mM), 0.1% (*w*/*v*) SDS (35 mM)) containing protease and phosphatase inhibitors (Complete Mini, Roche), in order to achieve the cleavage of the cell membranes. The total cellular extracts of the proteins were collected and the protein levels in each sample were measured by using the Bradford method and run on SDS page electrophoresis as described elsewhere [69]. The intensity values from the densitometry analysis of Western blots were normalized against the corresponding loading control using ImageJ analysis software (NIH).

### 4.7. Folin–Ciocalteu Method for the Quantification of Total Phenolic Content

The total phenolic content of the extracts was determined using the Folin–Ciocalteu method as previously described [70] with a few modifications. Folin–Ciocalteu was used as the reagent and GA as the standard. Initially, a set of GA standard solutions at concentrations of 0.010, 0.025, 0.050, 0.100, 0.250 and 0.500 μg/mL were prepared and used to construct a calibration curve. 100 μL of extract (concentration 2 mg/mL) or standard solution was added to a vial. Subsequently, 4.5 mL of distilled water and 100 μL Folin–Ciocalteu reagent were added, and the vial was shaken vigorously. After 3 min, 300 μL of Na_2_CO_3_ (2%) solution was added and the mixture was allowed to react for 2 h shaking occasionally. The absorbance was measured at 760 nm on a UV spectrophotometer (UV-1700 series, Shimadzu, Kyoto, Japan). The total phenolic content was expressed as GA equivalents (mg GAE/g extract).

The linearity of standard calibration curve was evaluated by linear regression analysis. The standard calibration curve was calculated by the least squares regression method to calculate the calibration equation and the determination coefficient (R^2^). Figure 8 shows the standard calibration curve of the F–C method. A linear correlation was found between absorbance of the blue complex at 760 nm and concentration of gallic acid in the range 0.010–0.500 mg/mL. The determination coefficient obtained from the linear regression was 0.9999, indicating excellent linear correlation between the data. From the slope of the calibration curve, LOD and LOQ were established. The LOD and LOQ for the F-C method were 0.0055 and 0.166 mg GAE/mL, respectively.

### 4.8. DPPH (2,2-Diphenyl-1-picryl-hydrazyl-hydrate) Radical Scavenging Assay

The DPPH radical scavenging assay was performed according to the protocol reported by Sarikurkcu et al. (2009) [71] with some modifications. 1 mL of each extract solution of concentrations 0.001, 0.002, 0.005, 0.01, 0.05, 0.1, 0.5 and 1 mg/mL were prepared and added to a 1 mL of DPPH fresh radical solution in methanol (0.2 mM). A control sample was prepared, which contained 1 mL H_2_O and 1 mL DPPH. After the mixtures were shaken vigorously and left in the dark for 30 min at room temperature, the absorbance was measured at 517 nm (Shimadzu, UV-1700). Radical scavenging capacity was expressed as a percentage effect (I%) and calculated using the following equation:(2)$$I\;\left(\%\right)=100\;\times \frac{A_{control}-\;A_{sample}}{A_{control}}$$
where *A_control_* is the absorbance of the control and *A_sample_* is the absorbance of the extract.

Antiradical curves were plotted referring to concentration on the x axis and their relative scavenging capacity on the y axis. The EC_50_ values were calculated using the Origin program.

### 4.9. LC-MS Analysis

Analyte identification criteria were relative retention time within ±0.05 min of the mean calibrator retention time and SIM mode of parent ion of each compound. For quantitative determination, calibration curves were constructed by plotting the peak areas of polyphenols against their concentrations, which were in the range 1–15 ng/mL for Apigenin, Gallic acid and Ferulic acid, 3–25 ng/mL for Rutin and Kaempferol, 10–50 ng/mL for Naringenin and 15–100 ng/mL for Myricetin. The obtained regression equations revealed a linear relationship between the peak area of each polyphenol species and their concentration, with corresponding correlation coefficients higher than 0.995. The limit of detection (LOD) and limit of quantification (LOQ) were calculated based on the standard deviation of the response and the slope (Appendix A). The detection limits were expressed as:LOD = 3.3 σ/Slope, LOQ = 10 σ/Slope(3)
where: σ = the standard deviation of the response at low concentrations. Slope = the slope of the calibration curve.

DE and EA ripe pulp + seeds extracts were dissolved in methanol (LC-MS grade) to obtain the final concentration of 250 ppb, sonicated and filtered through 0.45 μm PTFE syringe filters prior to analysis.

HPLC-MS analysis was performed according to a previously reported method with some modifications [72]. Analyses were performed using an Agilent Technologies Infinity liquid 1260 chromatographic system, consisting of an autosampler, a column thermostat and a binary solvent management system. MS data were acquired via positive electrospray ionization (ESI) in SIM mode. The chromatographic separation was accomplished using a Venusil XBP C18 column (150 × 4.6 mm, 5 μm) with a pre-column of the same material. A binary A/B gradient elution was used with solvent A: Milli Q water with 0.01% TFA, and solvent B: ACN. The gradient program was as follows: initial conditions were 100% A, raised to 98% B over the first 30 min. For re-equilibration of the system before the next injection, 15 min post time was used. Ten microliters were injected into the system with a flow rate of 0.8 mL/min and column temperature at 25 °C.

### 4.10. Statistical Analysis

Results for continuous variables were presented as Mean Standard Error. Two-group differences in continuous variables were assessed by the unpaired *t*-test. P-values are two-tailed with confidence intervals 95%. Statistical analysis was performed by comparing treated samples with untreated control. All statistical tests were conducted using Prism software version 8.0 (Graphpad, San Diego, CA, USA).

## 5. Conclusions

These results suggest that the extracts derived from the Cypriot carob fruit may show anti-cancer activity against breast cancer cells while normal cells remain unaffected. The extracts produced from the whole fruit with Diethyl ether and Ethyl acetate were found to be most effective in inhibiting proliferation and inducing apoptosis selectively in MCF-7 breast cancer cells. Overall, the extracts that exhibited the strongest anti-oxidant capacity had the highest TPC. The dominant polyphenols in the most potent extracts were found to be Myricetin, Naringenin and Kaempferol, which have been previously reported to have anti-cancer and health promoting effects. The proposed mechanism of action of the effective carob extracts involves activation of the intrinsic pathway of apoptosis and inhibition of cell proliferation (Graphical Abstract). These effects may be attributed to the phenolic content of the extracts which varies based on the extraction solvent used, the stage of maturity as well as the part of the carob fruit involved.

## Figures and Tables

**Figure 1 molecules-26-05017-f001:**
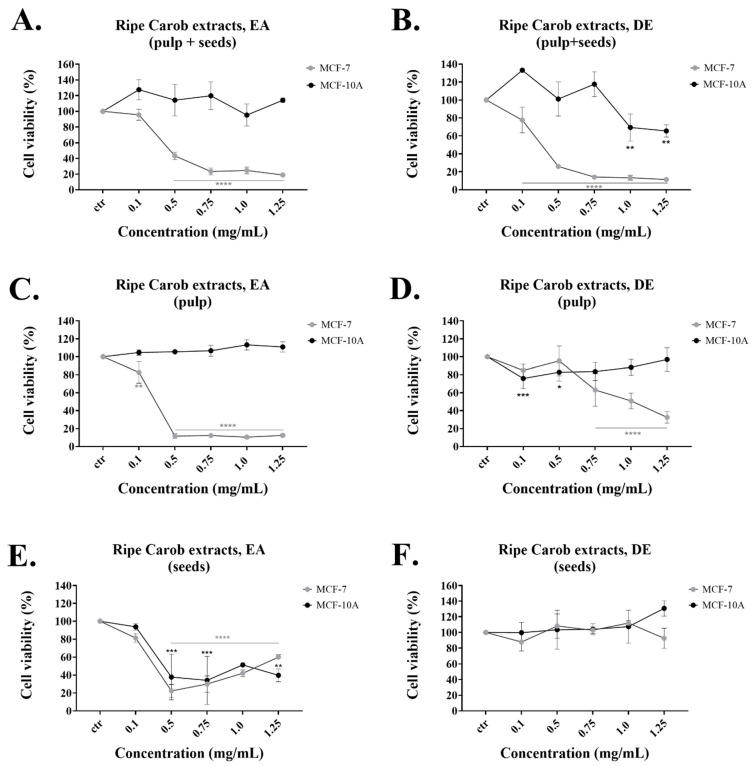
Antiproliferative effect of extracts produced from different solvents and parts of the ripe carob fruit. Comparison of the effect of pulp + seeds extract prepared with (**A**) EA or (**B**) DE, ripe pulp extract prepared with (**C**) EA and (**D**) DE and ripe seeds extract prepared with (**E**) EA and (**F**) DE on MCF-7 and MCF-10A cell viability at 48 h of treatment. All data are presented as mean values ± standard deviation and are representative of at least three independent experiments. *p* values: * <0.05, ** <0.01, *** <0.001, **** <0.0001 compared to control.

**Figure 2 molecules-26-05017-f002:**
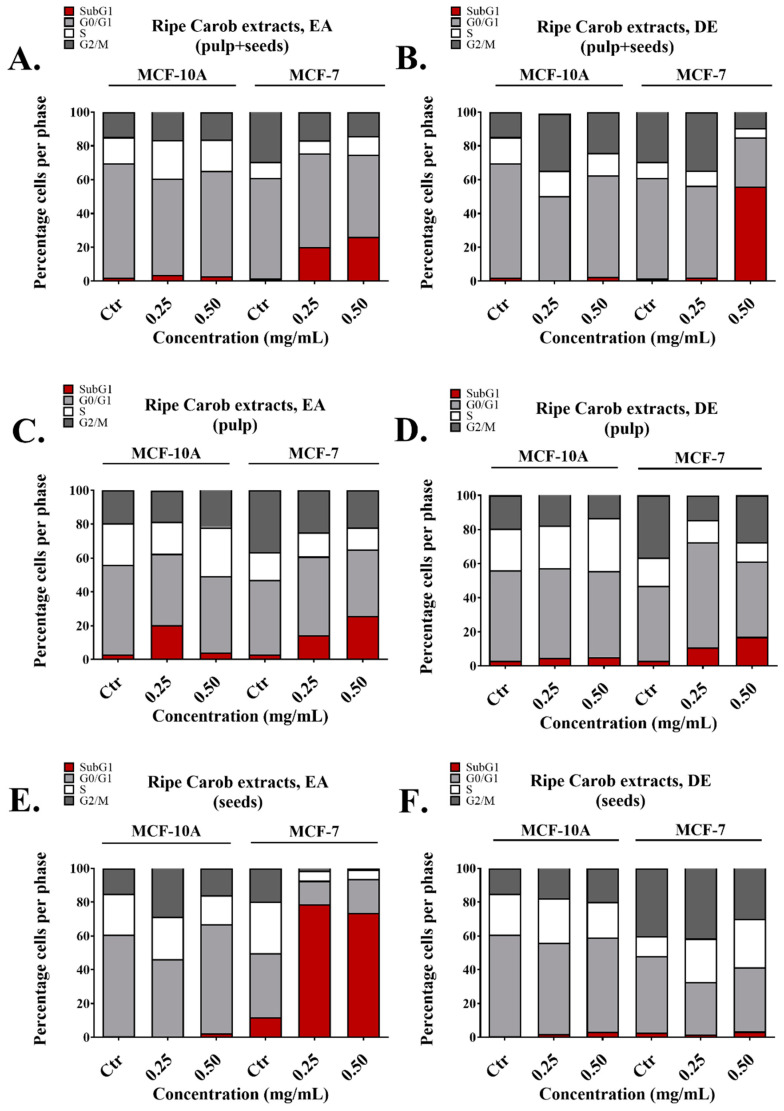
Cell cycle analysis following incubation of MCF-7 and MCF-10A cells with selected doses of Carob extracts for 48 h of treatment. Distribution of cell cycle phases following incubation with ripe pulp + seeds extracted with (**A**) EA, (**B**) DE, ripe pulp extracted with (**C**) EA, (**D**) DE, ripe seeds extracted with (**E**) EA and (**F**) DE. The results represent the mean SEM of at least three different replicates.

**Figure 3 molecules-26-05017-f003:**
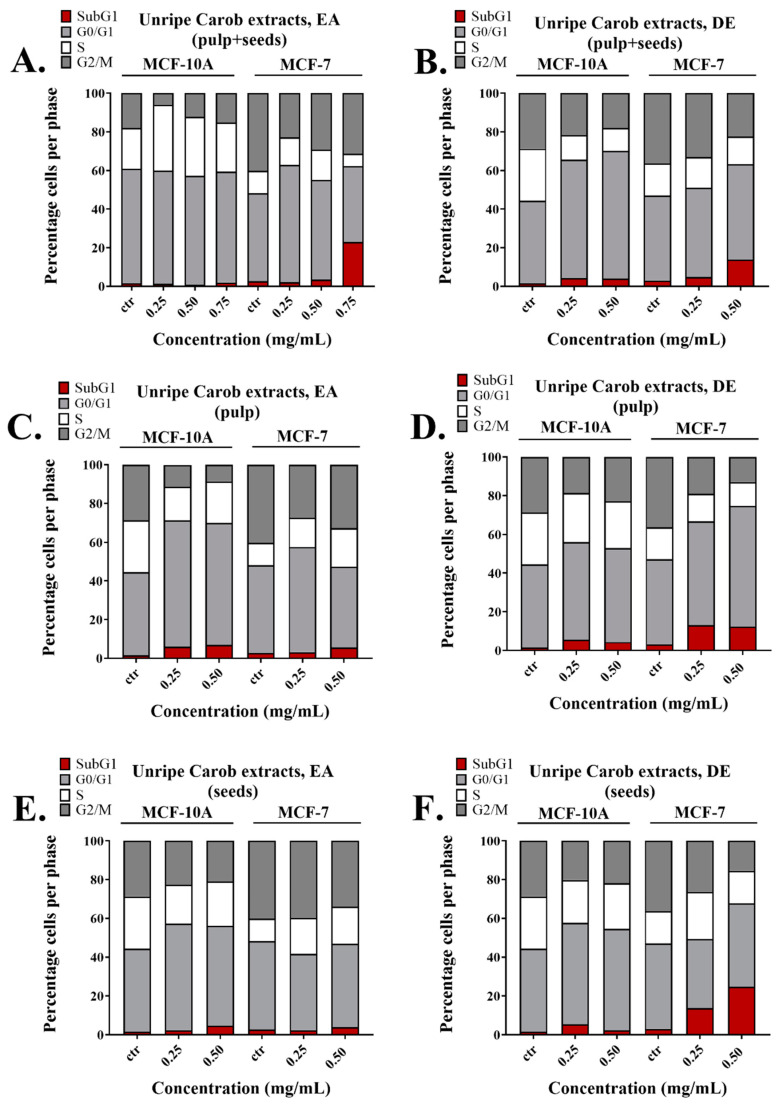
Comparison of the effects of unripe carob extracts on the cell cycle of MCF-7 and MCF-10A cells. Cell cycle analysis showing distribution of cell cycle phases following incubation of unripe pulp + seeds extracted with (**A**) EA, (**B**) DE, unripe pulp extracted with (**C**) EA and (**D**) DE, unripe seeds extracted with (**E**) EA and (**F**) DE. The results represent the mean SEM of at least three different replicates.

**Figure 4 molecules-26-05017-f004:**
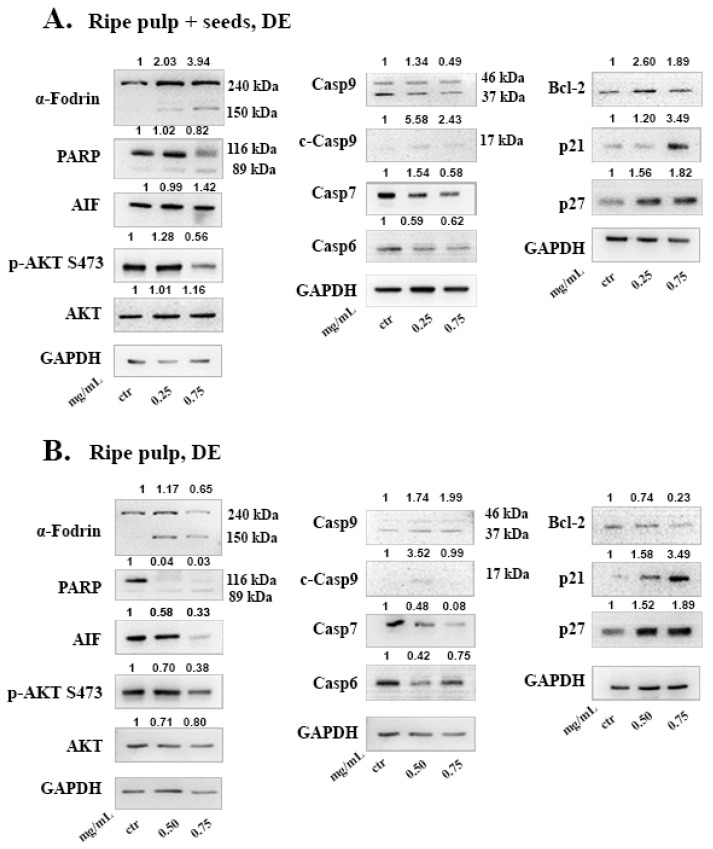
Effects of the Ripe pulp + seeds (**A**) and Ripe pulp (**B**) carob extracts prepared with DE on apoptotic proteins. Cells were treated for 48 h and 40 μg of protein was loaded. The numbers above the images represent the densitometry values compared to untreated control and normalized against GAPDH. Results are representative of at least three repetitions.

**Figure 5 molecules-26-05017-f005:**
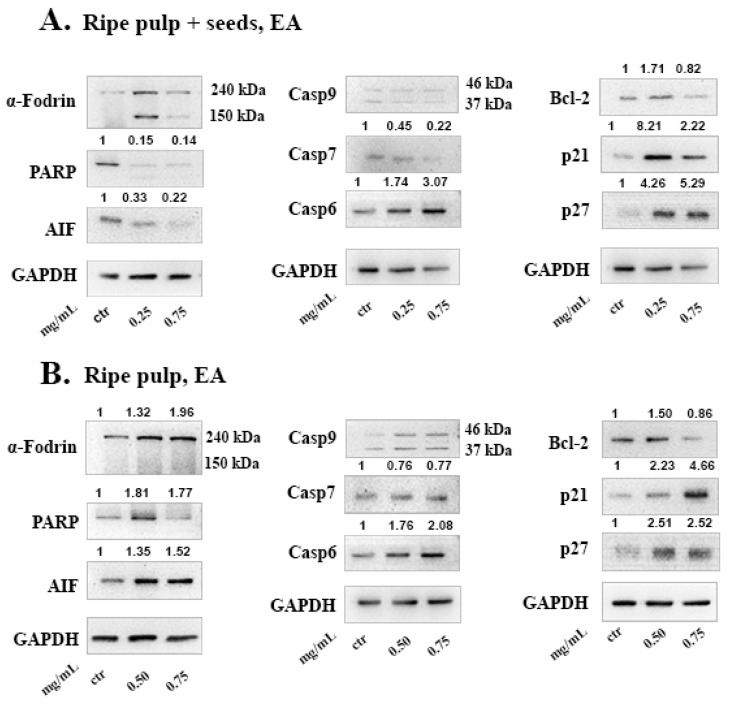
Effects of the EA ripe carob extracts in apoptosis induction. (**A**) EA ripe pulp + seeds extract. (**B**) EA ripe pulp extract. Cells were treated for 48 h and 40 μg of protein was loaded. The numbers above the images represent the densitometry values compared to untreated control and normalized against GAPDH. Results are representative of at least three repetitions.

**Figure 6 molecules-26-05017-f006:**
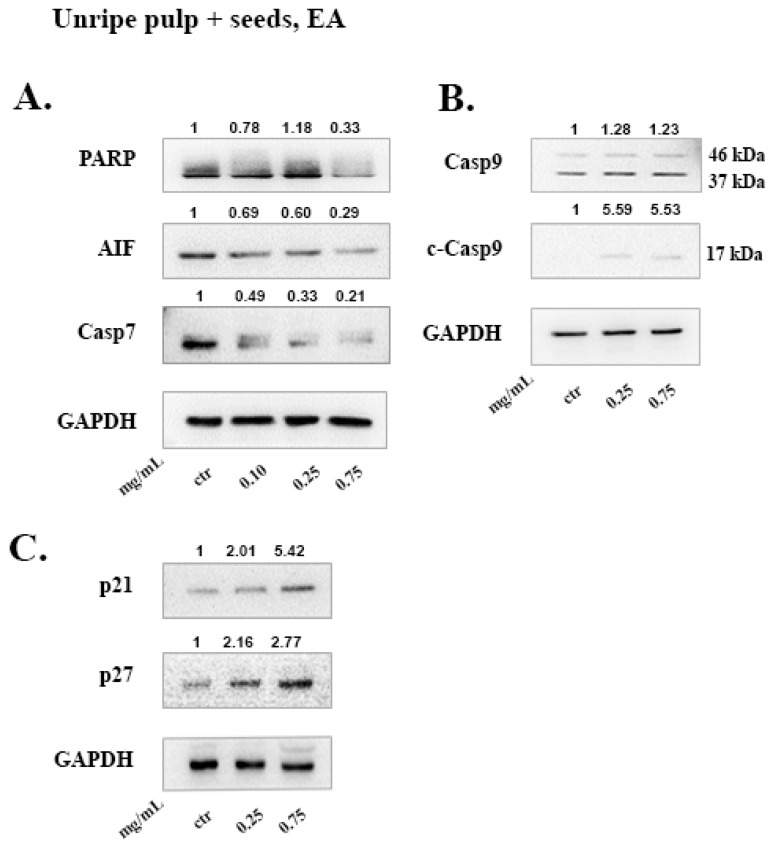
Effects of the unripe extract in caspase-dependent and independent apoptosis induction. EA unripe pulp + seeds extract induced (**A**) decreased PARP, AIF and caspase-7 levels, (**B**) the activation of caspase-9 and (**C**) increased p21 and p27 protein expression. 48h of treatment was performed and 40 μg of protein was loaded, the numbers represent the densitometry values compared to control (GAPDH). Results are representative of at least three repetitions.

**Figure 7 molecules-26-05017-f007:**
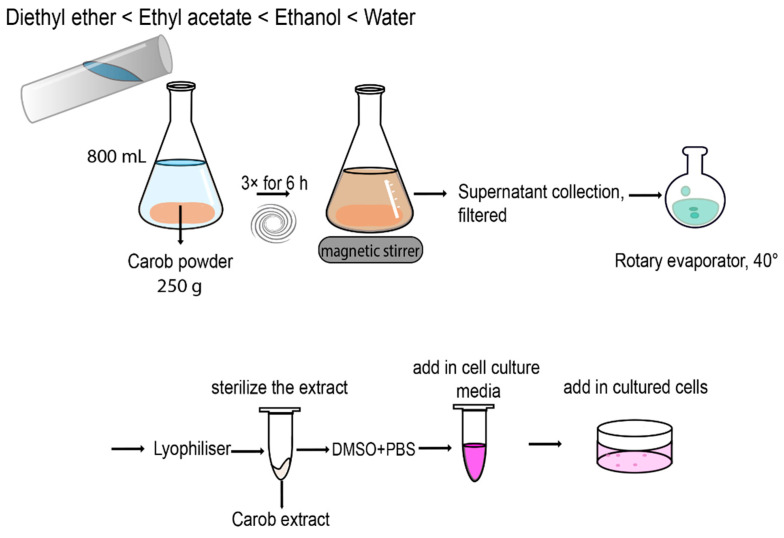
Schematic illustration for the preparation of carob extracts.

**Figure 8 molecules-26-05017-f008:**
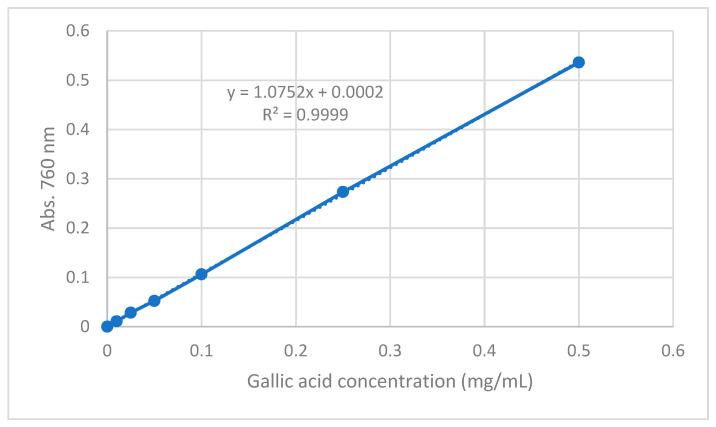
Gallic Acid calibration curve for Folin–Ciocalteu method.

**Table 1 molecules-26-05017-t001:** Half maximal inhibitory concentration (IC_50_) of the Ripe pulp + seeds Carob Extracts. The concentration is marked as N/A, (Not Available), in the cases where viability did not reach 50%.

IC_50_ (mg/mL) of Ripe Carob Extracts		
Pulp + Seeds		
Solvent	MCF-7	MCF-10A
Ethyl Acetate	0.44 ± 0.05	N/A
Diethyl Ether	0.26 ± 0.03	N/A
Ethanol	0.76	5.43 ± 0.66
Aqueous	2.18 ± 0.22	5.74 ± 1.17

(N/A, Not Available; IC_50_, half maximal inhibitory concentration).

**Table 2 molecules-26-05017-t002:** Half maximal inhibitory concentration (IC_50_) of the ripe pulp or seeds Carob Extracts. The concentration is marked as N/A (Not Available), in the cases where viability did not reach 50%.

IC_50_ (mg/mL) of Ripe Carob Extracts				
	Pulp		Seeds	
Solvent	MCF-7	MCF-10A	MCF-7	MCF-10A
Ethyl Acetate	0.27 ± 0.03	N/A	0.26 ± 0.02	0.57 ± 0.39
Diethyl Ether	0.96 ± 0.11	2.34 ± 0.03	N/A	N/A

(N/A, Not Available; IC_50_, half maximal inhibitory concentration).

**Table 3 molecules-26-05017-t003:** Half maximal inhibitory concentration (IC_50_) of the unripe Carob extracts. The concentration is marked as N/A, (Not Available), in the cases where viability did not reach 50%. The data are expressed as the mean (±) of the results from at least three different experiments.

IC_50_ (mg/mL) of Unripe Carob Extracts						
	Pulp + Seeds		Pulp		Seeds	
Solvent	MCF-7	MCF-10A	MCF-7	MCF-10A	MCF-7	MCF-10A
Ethyl Acetate	0.82 ± 0.08	N/A	1.45 ± 0.11	0.29 ± 0.02	N/A	N/A
Diethyl Ether	0.42 ± 0.20	1.75 ± 0.19	0.28 ± 0.06	0.38 ± 0.09	N/A	N/A

(N/A, Not Available; IC_50_, half maximal inhibitory concentration).

**Table 4 molecules-26-05017-t004:** Displaying TPC and EC_50_ values of the Carob extracts. ***** EC_50_: concentration required to obtain a 50% anti-oxidant effect as measured by the inhibition of the DPPH radical scavenging activity.

Extract	Total Phenolic Content (TPC) (mg/g)	EC_50_ * (μg/mL)
EA ripe pulp + seeds	37.99	63.11
EA ripe pulp	11.57	165.08
EA ripe seeds	14.74	104.89
DE ripe pulp + seeds	10.71	450.56
DE ripe pulp	6.2	1618.70
DE ripe seeds	n.d.	n.d.
EA unripe pulp + seeds	140.41	11.51
EA unripe pulp	112.15	20.13
EA unripe seeds	n.d.	622.09
DE unripe pulp + seeds	12.42	304.32
DE unripe pulp	42.39	71.30
DE unripe seeds	n.d.	n.d.

(n.d.; not detected).

**Table 5 molecules-26-05017-t005:** Concentration of polyphenols found in EA and DE ripe pulp + seeds carob extracts (mg/mL extract, N = 3).

Compound	EA Extract ± SD	DE Extract ± SD
Apigenin	0.68 ± 0.02	0.92 ± 0.05
Myricetin	40.67 ± 0.37	53.76 ± 0.46
Rutin	3.30 ± 0.10	1.98 ± 0.02
Naringenin	19.20 ± 0.39	19.50 ± 0.22
Ferulic acid	0.56± 0.02	DNQ **
Kaempferol	2.65 ± 0.11	14.93 ± 0.28
Gallic acid	1.43 ± 0.01	1.46 ± 0.00
Sum	68.49	92.8

** DNQ: detected not quantifiable.

**Table 6 molecules-26-05017-t006:** Yield of extraction of ripe carob whole fruit and its parts with various solvents.

Solvent	Ripe Carob Part	Extraction Yield (%)
Ethyl Acetate	Pulp + Seeds	0.26
	Pulp	0.10
	Seeds	0.30
Diethyl Ether	Pulp + Seeds	0.34
	Pulp	0.06
	Seeds	1.32
Ethanol	Pulp + Seeds	17.18
Water	Pulp + Seeds	28.98

**Table 7 molecules-26-05017-t007:** Yield of extraction of the unripe whole carobs and its parts using DE and EA as shown.

Solvent	Unripe Carob Part	Extraction Yield (%)
Ethyl Acetate	Pulp + Seeds	0.09
	Pulp	0.12
	Seeds	0.8
Diethyl Ether	Pulp + Seeds	0.13
	Pulp	0.04
	Seeds	0.29

## Data Availability

The data presented in this study are available on request from the corresponding author.

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
