# Peer review of "Anti-Cancer Activity and Phenolic Content of Extracts Derived from Cypriot Carob (Ceratonia siliqua L.) Pods Using Different Solvents"

_molecules, 2021, doi:10.3390/molecules26165017_

Round 1

Reviewer 1 Report

The article shows a biological study of the species Ceratonia Siliqua. Although the study is important, the article has shortcomings, especially in methodologies

1)    The National Cancer Institute’s screening program indicates that crude extracts yielding IC50 values < 30 µg/mL are candidates for purification. The results presented in the article for the extract and fractions are way above these 30 µg/mL. How can the authors affirm that extract and fractions have activity? The lowest IC50 value was for the 260 µg/mL ethyl acetate fraction. This value is still almost 9 times higher than the value established by the NCI.

2)    In table 5 the authors show quantification data for the compounds, but they do not show standard deviation data. 
3)    Authors need to show figures of merit for compound quantification.

4)    In line 283, “LC-MS analysis of DE and EA ripe pulp+seeds carob extracts was performed in order”, but it does present no chromatogram information. n the figure S3, the authors show only the quantitation chromatograms

Author Response

1) The National Cancer Institute’s screening program indicates that crude extracts yielding IC50 values < 30 µg/mL are candidates for purification. The results presented in the article for the extract and fractions are way above these 30 µg/mL. How can the authors affirm that extract and fractions have activity? The lowest IC50 value was for the 260 µg/mL ethyl acetate fraction. This value is still almost 9 times higher than the value established by the NCI.

Our aim was to explore for the first time the potential anti-cancer action of extracts derived from Cypriot carobs. Following the Reviewer’s comment, we added lines 335-340 and 344-348 in the “Discussion” section to reference the IC50 values indicated by the US National Cancer Institute (NCI) screening program and to discuss the relatively high IC50 values of the carob extracts. Based on NCI guidelines, a crude extract is generally considered to have high in vitro cytotoxic activity if the IC50 value is ≤ 30 μg/mL (ref 34,35). Pezzuto et al consider extracts with higher IC50 values (100 μg/ml) to be cytotoxic and eligible candidates for further studies (ref 36), while several recently published papers (ref 37-42) explore the biological activity of plant extracts with IC50 values higher than 200 μg/ml. We showed that selected carob extracts not only affect the viability/proliferation of cancer cells but also induce apoptosis and changes in protein expression. In addition, the DPPH radical scavenging activity assay showed that the extracts had anti-oxidant effects with EC50 values starting from 11.51 μg/ml (Table 4). It is also important to note that even though the selected carob extracts may be considered to have moderate or low toxicity against cancer cells according to NCI guidelines, they were non-toxic to the normal cell line as viability did not reach 50% (Table 1). For this reason, we believe that future investigations should include the sub-fractionation of crude carob extracts in an effort to improve their IC50 values against cancer cells and screening of the crude extracts and their sub-fractions against a panel of cancer and normal cell lines to evaluate their effectiveness in a range of different cancer types.

2)    In table 5 the authors show quantification data for the compounds, but they do not show standard deviation data. 

The SD values were included in this revised manuscript. Please see Table 5.

3)    Authors need to show figures of merit for compound quantification.

Figures of merit, such as LODs, LOQs, regression equations, coefficients of determination, are demonstrated in Table S1. In this revised manuscript, precision, which was evaluated in terms of RSD%, was included. The RSD% values are demonstrated in Table S1.

4)    In line 283, “LC-MS analysis of DE and EA ripe pulp+seeds carob extracts was performed in order”, but it does present no chromatogram information. In the figure S3, the authors show only the quantitation chromatograms.

The Total Ion Chromatograms of the extracts are demonstrated in Figures S5 and S6 in this revised manuscript.

Reviewer 2 Report

The subject of the article is interesting and will undoubtedly interest many readers, not only specialists.  It is well written. All analytical methodologies were thoroughly characterized and the course of action was described. My comments concern:

The title does not reflect the actual issue of the conducted research. It is not clearly indicated that it is for testing with different solvents. Authors must change the title of the article.

The keywords remain the same as in the title of the article. So the authors need to change the keywords. They are an important part of the article on the basis of which you can find your article in search engines. However, they cannot and should not repeat the title of the work, and should be short and synthetic - key.

The introduction is too short. It does not reflect the subject matter of the problem. Line 73-84. Authors need to change. This is the methodical part that should be included in another chapter. Please indicate the main objectives of this study and emphasize the novelty of the already available published studies.

Author Response

1) The title does not reflect the actual issue of the conducted research. It is not clearly indicated that it is for testing with different solvents. Authors must change the title of the article.

We changed the title of the Manuscript to contain the information that different solvents were used for extraction.

2) The keywords remain the same as in the title of the article. So the authors need to change the keywords. They are an important part of the article on the basis of which you can find your article in search engines. However, they cannot and should not repeat the title of the work, and should be short and synthetic - key.

We replaced the keywords with shorter and more concise words so that most of them are no longer repeated in the title. 

3) The introduction is too short. It does not reflect the subject matter of the problem. Line 73-84. Authors need to change. This is the methodical part that should be included in another chapter. Please indicate the main objectives of this study and emphasize the novelty of the already available published studies.

We thank the Reviewer for the valuable feedback. Our aim was to explore for the first time the potential anti-cancer action of extracts derived from Cypriot carob pods. Even though several recent papers focus on the chemical and biological properties of Cypriot carobs extracts, their potential anti-cancer effects and mechanism of action have not been explored. Our objectives were the following: 1. To compare the efficacy of carob pod extracts derived using solvents of different polarities, known to affect polyphenol content, 2. To evaluate extracts derived from pods at different maturity stages (green and ripe) and 3. To compare the effects of extracts derived from different parts of the fruit (whole pod, seeds and pulp). We believe that the information contained in lines 73-84 of the unrevised manuscript should remain in the “Introduction” of the paper, so that the reader better appreciates the purpose of the work and why it is significant. Following the Reviewer’s recommendation, we have expanded on the “Introduction” section including the most significant results of recent studies. In addition, we emphasized our main objectives and the novelty of our work compared to available published studies (lines 76-89).

Reviewer 3 Report

The authors submitted a revised version of the manuscript in which all the suggested raccomandations were addressed. Consequently, I think that the manuscript is now suitable for the publication in molecules. 

Author Response

We thank the reviewer for the valuable feedback.

Reviewer 4 Report

This paper reports the antitumor effect of the extracts of Cypriot carob pods against MCF-7 and MCF-10A cells.

Comments are as follows.

#1. Some polyphenols (myricetin, naringenin, and so on) have already been previously reported to induce apoptosis in cancer cells. However, it is not clear to what extent the antitumor effect of phenols contributes to the ethyl acetate (EA) and diethyl ether (DE) extract in this paper. Those extracts should be separated by column chromatography or the other method into polyphenol-rich fraction and other fractions. These separated subfractions are needed to be evaluated respectively against antitumor activity (MTT assay, cell cycle analysis, Western blotting, and so on).

#2. The ethyl acetate (EA) extract from ripe carob pulp seems to have most potent antitumor activity in the Figure 1. This extract shows selectivity activity for inhibiting the proliferation of cancer cells, and does not contain seeds extract which has cytotoxicity against normal cells. The authors should describe the reason why the statement “The extracts produced from the whole fruit with Diethyl ether and Ethyl acetate were found to be most effective in inhibiting proliferation and inducing apoptosis selectively in MCF-7 breast cancer cells.” (lines 595-598) is conclusively presented. It is unlikely that the Figure 1 support this statement.

Author Response

1) Some polyphenols (myricetin, naringenin, and so on) have already been previously reported to induce apoptosis in cancer cells. However, it is not clear to what extent the antitumor effect of phenols contributes to the ethyl acetate (EA) and diethyl ether (DE) extract in this paper. Those extracts should be separated by column chromatography or the other method into polyphenol-rich fraction and other fractions. These separated subfractions are needed to be evaluated respectively against antitumor activity (MTT assay, cell cycle analysis, Western blotting, and so on).

We thank the Reviewer for the valuable feedback. Our aim was to explore for the first time the potential anti-cancer action of crude extracts derived from Cypriot carob pods and perform LC-MS analysis to identify the most abundant polyphenols in the extracts with the most interesting anti-cancer action. For this reason, we first investigated the antiproliferative capacity of 14 different crude extracts using the MTT assay, and then focused on selected EA and DE extracts to thoroughly investigate their anti-cancer effects by performing cell cycle analysis, Western blot and the DPPH radical scavenging activity assay. We agree with the Reviewer that the observed antitumor effects of the DE and EA extracts cannot be definitively attributed solely to the presence of the identified polyphenols, and that sub-fractionation of crude carob extracts followed by extensive evaluation for anticancer effects would be of great value. This would not only clarify which of the constituents are responsible for the observed effects but may also potentially lead to improved efficacy against cancer cells. Based on the Reviewer’s comment, we have added lines 426-431 in the “Discussion” section stressing the importance of sub-fractionation and further investigation of antitumor effects in future studies. This is also mentioned in lines 359-363.

2) The ethyl acetate (EA) extract from ripe carob pulp seems to have most potent antitumor activity in the Figure 1. This extract shows selectivity activity for inhibiting the proliferation of cancer cells, and does not contain seeds extract which has cytotoxicity against normal cells. The authors should describe the reason why the statement “The extracts produced from the whole fruit with Diethyl ether and Ethyl acetate were found to be most effective in inhibiting proliferation and inducing apoptosis selectively in MCF-7 breast cancer cells.” (lines 595-598) is conclusively presented. It is unlikely that the Figure 1 support this statement.

The antiproliferative activity of EA extract from ripe carob pulp as shown in Figure 1 and based on the IC50 values of Table 2 (0.27 mg/ml) is indeed comparatively potent compared to DE and EA extracts from the whole fruit (IC50 values 0.26 and 0.44 mg/ml respectively, Table 1). However, the statement made in lines 595-598 of the unrevised version of the manuscript, refers not only to the antiproliferative potency but also to the selective apoptosis induction as observed in Figure 2.  Cell cycle analysis revealed that the Diethyl ether and Ethyl acetate whole fruit extracts increased the subG1 fraction (indicative of apoptosis induction) in MCF-7 cancer cells only (Fig. 2A, B). The EA extract from ripe carob pulp also induces apoptosis in normal immortalized MCF-10A cells as shown in Figure 2C. Following the Reviewer’s recommendation, we have added lines 357-358 in the “Discussion section”, to state that the selectivity of extracts against cancer cells was evaluated not only by the results of Figure 1 but also by their pro-apoptotic properties as shown in Figure 2.

Round 2

Reviewer 4 Report

Comments are as follows.

#1. Generally, the cytotoxic activity of the extracts can be determined by the MTT assay, PI staining, and so on. However, it is difficult to judge from this paper’s data that death of cells is based on apoptosis or necrosis. In Figure 1 and Figure 2, contribution of apoptosis (subG1) is shown only 20% in most cases. The rate of apoptosis and necrosis of each extract should be cleared by observing the cell morphology.

#2. The authors state “some extracts induced cleavage of caspase-9” (Page 9, 10, and 14), but from Western blot analysis results in Figure 4 and Figure 5, caspase-9 cleaved form (37 kDa) does not seem to increase. The authors’ above statement contradict these data in Figure 4 and Figure 5. The authors should confirm that Western blot analysis data reproducible.

Author Response

1. Generally, the cytotoxic activity of the extracts can be determined by the MTT assay, PI staining, and so on. However, it is difficult to judge from this paper’s data that death of cells is based on apoptosis or necrosis. In Figure 1 and Figure 2, contribution of apoptosis (subG1) is shown only 20% in most cases. The rate of apoptosis and necrosis of each extract should be cleared by observing the cell morphology.

The subG1 phase as measured by Flow Cytometry, is generally accepted as a valid measurement of the apoptotic cell population present in a sample. We have added relevant references that support this statement in this revised version of the paper (ref. 46, 47). As the Reviewer has pointed out, in some cases, the observed apoptosis as evaluated by the subG1 phase (Fig. 2), is inconsistent with the percentage drop in viability as measured by the MTT assay (Fig. 1). This may be due to the fact that the MTT assay measures cell metabolic activity while the appearance of the sub G1 phase during Flow Cytometry is attributed to the presence of DNA fragments. However, we agree with the Reviewer that other forms of cell death, including necrosis, may contribute to the observed effect of the carob extracts. Following the Reviewer’s recommendation, we added the importance of distinguishing between apoptotic and necrotic effects of the extracts in the “Discussion” section, lines 358-364.

  1. The authors state “some extracts induced cleavage of caspase-9” (Page 9, 10, and 14), but from Western blot analysis results in Figure 4 and Figure 5, caspase-9 cleaved form (37 kDa) does not seem to increase. The authors’ above statement contradict these data in Figure 4 and Figure 5. The authors should confirm that Western blot analysis data reproducible.

In our results, we showed that the DE ‘ripe pulp+seeds’ (Fig 4A), DE ‘ripe pulp’ (Fig. 4B) and DE ‘ripe pulp+seeds’ (Fig. 6) carob extracts induce the cleavage of caspase-9 by increasing the levels of the 17 kDa fragment (p17 subunit) following treatment. The levels of cleaved caspase-9 were measured by densitometry and are shown above each image, relative to the untreated control and the loading control (GAPDH levels). The p17 subunit, represents the cleaved form of caspase-9 without the CARD prodomain. We have added this information along with relevant references in lines 249-251.

This manuscript is a resubmission of an earlier submission. The following is a list of the peer review reports and author responses from that submission.

Round 1

Reviewer 1 Report

The article is interesting.
However, at the outset it is burdened with errors.
The title is unfortunately misleading. Because it is not a chemical composition, but only the total content of polyphenols. Therefore, the title should be reworded to reflect the actual content of the article and the experiments carried out. There is no information in the entire study which polyphenolic substances were identified as a result of the HPLC analysis of extracts. It would be good if the authors provided information on what polyphenols were dominant in these extracts and how the extracts obtained from different parts of plants differed. After all, they used different solvents. The chromatogram itself without description is only a graph. Therefore, the authors are asked to describe the chromatograms numerically or by inserting the names of the substances present in the various extracts. It seems somewhat optimistic to say that ... "The extracts made from the DE ripe seeds did not exert any anti-proliferative effects on either cell line. This is in agreement with our results from the HPLC analysis and other research studies, which reported that the carob seeds have lower amounts of polyphenols compared to the pulp ".... Because it is not known what the authors found by HPLC analysis as they did not report it in the results. Then, Figure 7.  HPLC chromatograms of phenolic compounds (extracted with DE) from A. 'pulp + 309 seeds', B. 'seeds' and C.' pulp '. The extract made from the 'pulp + seeds' displays the highest content 310 of polyphenols. - should be described so that it is clear which peak is a specific compound.

We also do not find this in the additional materials. The authors included a photo of TLC plates in them, but did not deign to describe them. Fig 4. in the file "molecules-1222061-original-images for Blots or Gels" - shows only the spectra, but we don't know what compounds have been identified. It is not known at what retention time the reference substances came out and what these were. The authors mention tannins, but it is also phenolic acids, flavonoids and many other substances that create the activity of the extracts. Also, if the authors discuss tannins, it would probably need to be labeled. Or at least determine which tannins can participate in the extracts and modulate anti-cancer activity.
Therefore, I suggest changing the abstract, as it does not reflect the actual information from the experiment, especially in the statement: ..."Finally, we used HPLC analysis to identify and quantify polyphenols in the most effective extracts. Our results demonstrate that the anti-proliferative capacity of carob extracts varied with the stage of carob maturity and the solvent extraction. The Diethyl ether extraction and Ethyl acetate extracts derived from the ripe whole fruit had the highest Gallic acid content and also displayed specific activity against cancer cells" ... I don't know where in the article the authors confirm this information?

The summary is laconic and overly optimistic. It does not include the most important points from the experiments performed.
The literature is largely quite old. They should be refreshed with reports from the last 10 years. Such literature is publicly available.

Reviewer 2 Report

The manuscript entitled “Anti-cancer Activity and Chemical Composition of Extracts Derived from Cypriot Carobs”, authored by Gregoria Gregoriou and colleagues, deals with the investigation of the anti-cancer properties of carob extracts on cancer and normal immortalized breast cells via MTT assay, cell cycle analysis and Western Blotting. Moreover, the authors evaluated total polyphenol content and anti-oxidant property via Folin-Ciocalteu method and the DPPH assay. The article is really complete, interesting and well written. In particular, I find it written with a lot of authority.

I would like to give just a few small suggestions to the authors:

Keywords: Keywords should be words not contained in the title, at most present in the abstract. Their usefulness is to make easier the searching of the article using the common scientific search engines. Since several keywords are already present in the title, and/or repeated several times in the abstract, I strongly advise the authors to change some of the proposed keywords with other news. As author guidelines clearly report, authors can provide up to 10 different keywords.

Introduction:

  1. Authors stated that carob fruits containing a large variety of bioactive compounds, that exert different ‘pharmacological’ actions, including antiproliferative. However, this topic is not exclusive to the fruit under evaluation of the authors, but several fruits, leaves, and roots have reported similar effects. The authors should specify better in the introduction that the consumption of plant materials could have effects different from that expected (satiety). This is due to the fact that plant material, whether edible or not, are very rich sources of bioactive compounds that can carry out these ‘pharmacological’ actions. In this context, very recently various extracts from leaves (10.3390/molecules25112612), seeds (10.1016/j.foodchem.2019.125909), peel (10.3390/molecules26030564) of edible plants have been extensively studied for their antiproliferative action.
  2. Moreover, I think that the topic treated by the authors falls within a certain point of view in the valorisation of waste products from food processing. In particular, since the seeds of the fruits are discarded during the production of juices, yoghurt, jams and other food products, they could constitute a potential environmental contamination. The valorisation of these waste products for the production of dietary supplements (3390/molecules25112612; 10.1038/s41598-020-60252-7), plant biostimulant (doi.org/10.3390/plants9101308; doi.org/10.1038/s41598-020-79770-5) or other typologies of products (doi.org/10.1021/acssuschemeng.9b07479) could therefore be a strong point of this article.

Results:

  • the acronyms listed in table 1 should be reported as footnotes.
  • In table 4, the values ‘0’ should be replaced with n.d. (not detected). Moreover, in order to be able to state the absence of a compound, the authors should report the calculation of the LOD and LOQ for the spectrophotometric values ​​in the materials and methods section.

Reviewer 3 Report

The article shows a biological study of the species Ceratonia Siliqua. The results are very interesting and the article brings a lot of consistent data that show the potential of the species but considers that some necessary corrections must be made before being accepted to publish.

  • The name of the investigated species must be presented in the title of the manuscript;

       2) The authors present in Figure 7, the chromatographic profile of the extract extracted by HPLC, but it does not present any information on UV spectra and chemical characterization;

I recommend an LC-MS analysis to show the chemical composition of the most active extracts;

      3) In line 313, the authors cite a quantification of substances and retention times of major compounds. How did the authors carry out this quantification? The authors do not show the validation data for the quantification method;

     4) In line 544, the authors cite the LC-MS methodology, but it does not show any results of the analysis. I recommend showing the chromatogram and mass fragments for all chromatographic peaks.